# Developing a Model for Integrating of Tuberculosis, Human Immunodeficiency Virus and Primary Healthcare Services in Oliver Reginald (O.R) Tambo District, Eastern Cape, South Africa

**DOI:** 10.3390/ijerph20115977

**Published:** 2023-05-28

**Authors:** Ntandazo Dlatu, Benjamin Longo-Mbenza, Kelechi Elizabeth Oladimeji, Teke Apalata

**Affiliations:** 1Department of Public Health, Faculty of Health Sciences, Walter Sisulu University, Private Bag X1, Mthatha 5117, South Africa; longombenza@gmail.com; 2College of Graduate Studies, University of South Africa, Johannesburg 0001, South Africa; oladimejikelechi@yahoo.com; 3Department of Laboratory Medicine and Pathology, Faculty of Health Sciences and National Health Laboratory Services (NHLS), Walter Sisulu University, Private Bag X1, Mthatha 5117, South Africa; ruffinapalata@gmail.com

**Keywords:** model development, TB–HIV integrated model, TB and HIV, model, quantitative and qualitative data

## Abstract

Despite the policy, frameworks for integration exist; integration of TB and HIV services is far from ideal in many resource-limited countries, including South Africa. Few studies have examined the advantages and disadvantages of integrated TB and HIV care in public health facilities, and even fewer have proposed conceptual models for proven integration. This study aims to fill this vacuum by describing the development of a paradigm for integrating TB, HIV, and patient services in a single facility and highlights the importance of TB-HIV services for greater accessibility under one roof. Development of the proposed model occurred in several phases that included assessment of the existing integration model for TB-HIV and synthesis of quantitative and qualitative data from the study sites, which were selected public health facilities in rural and peri-urban areas in the Oliver Reginald (O.R.) Tambo District Municipality in the Eastern Cape, South Africa. Secondary data on clinical outcomes from 2009–2013 TB–HIV were obtained from various sources for the quantitative analysis of Part 1. Qualitative data included focus group discussions with patients and healthcare workers, which were analyzed thematically in Parts 2 and 3. The development of a potentially better model and the validation of this model shows that the district health system was strengthened by the guiding principles of the model, which placed a strong emphasis on inputs, processes, outcomes, and integration effects. The model is adaptable to different healthcare delivery systems but requires the support of patients, providers (professionals and institutions), payers, and policymakers to be successful.

## 1. Introduction

Although great progress has been made over the years in the control of both epidemics, tuberculosis (TB) remains one of the main causes of mortality and poor health among people living with human immunodeficiency virus (HIV), especially in low-resource countries [1]. In fact, the spread of HIV in sub-Saharan Africa has significantly increased the incidence of TB [2]. There is strong evidence that HIV and TB are related, with higher mortality rates among co-infected patients [3,4]. Programs at the global, national, and local levels began primarily with a vertical approach with little or no coordination [5,6,7], despite a significant risk of HIV patients developing TB. For this reason, syndemic disease was poorly managed, negatively impacting clients and creating operational challenges for service providers, particularly in resource-poor settings [8]. The World Health Organization (WHO) proposed TB-HIV to integrate services at least at the facility level in response to persistent TB-HIV comorbidity [9]. In full integration, human immunodeficiency virus treatment and TB are provided by the same-trained health care provider at the same visit, a ‘one-stop service’: TB provides HIV treatment, and the HIV clinic provides TB treatment. This model is considered the most efficient and effective way to provide comprehensive TB-HIV services, and it is appropriate for facilities with high TB and HIV prevalence [4]. However, there is disagreement about the degree of integration (full or partial) and the levels at which it should occur [8,9,10]. As a result, numerous models (linkage, collaboration, full integration) have been put into practice in different situations with a number of difficulties [11,12,13,14]. According to the collaborative model WHO, a person who has received a TB diagnosis also receives counseling and an HIV test, and if the test is positive, he or she is referred [3]. All TB and HIV services are provided in one location by the same service providers under a fully integrated model of care. Evidence suggests that this fully integrated option offers the best benefits for patients, health systems, and staff [14,15,16,17]. Regardless of the documented intention to fully integrate [18,19,20,21], widespread integration of TB-HIV care remains unsatisfactory in many resource-limited settings such as South Africa, where TB remains a public health challenge largely due to the high HIV prevalence estimated at 12% in the general population [22,23,24,25]. South Africa has opted for the integrated model, in which TB and HIV services are provided in a single facility at the same time and place. This is a potentially effective model for SA to address the high prevalence of HIV, TB, and HIV-TB coinfections, as well as limited human resources [4]. Although this is not the first empirical discussion of TB-HIV integration in South Africa, it is, as far as we know, the first attempt to examine operational issues of TB-HIV integration from the perspective of nurses at the facility level [26,27,28,29,30,31]. This paper describes the development of a district-based model for comprehensive and integrated TB and HIV care that meets the needs of the region. At the time of writing, health services in South Africa were still provided separately, with few formal linkages between them. Although referral systems are improving, it is still difficult to track patients sent outside a clinic, and patients still miss the opportunity to receive comprehensive TB and HIV care. This is true even in health facilities with limited resources, such as too few staff, too little money, and a high volume of TB and/or HIV-infected patients.

## 2. Study Aim and Setting

We aimed to promote the scaling up of TB and HIV services, highlight the challenges of delivering integrated TB-HIV services at scale, and contribute to the development of integrated services policy in South Africa through the design of an improved integration model based on empirical data evaluation. The study population were patients aged 18 years of age or older in the study setting, which also included five selected public healthcare facilities in both rural and peri-urban settings in the Oliver Reginald (O.R.) Tambo District Municipality in the Eastern Cape, South Africa. In addition to being one of the poorest districts within the Eastern Cape is one of the ten districts with the greatest combined burden of HIV and TB [32,33]. The country has estimated prevalence of 737/100,000 was higher than the Eastern Cape’s incidence of TB in 2017 of 839/100,000 [32,33]. Additionally, the Eastern Cape had a high prevalence rate of HIV overall of 25.2% as of 2017 [19,32,33]. In order to improve both the TB and HIV care given at public healthcare settings, in terms of services provided to patients where they dwell, we set out to research cutting-edge approaches. In this article, we discuss the work that was done to help with the development of the integration model (study design), give a brief summary of the baseline data that was used to inform the model (key findings from baseline research findings), describe how the model was developed and its main components (discussion: designing the model), and describe how the model was implemented. The integration model’s efforts for developing health care systems for the delivery of TB and HIV integrated care rely on straightforward, tried-and-true techniques as well as cutting-edge, novel concepts.

## 3. Methods

### 3.1. Part 1

For quantitative assessment to inform the development of the model in part 1 phase, secondary data on TB-HIV clinical outcomes was obtained from a range of sources, including governmental, non-governmental, and research institutes [32]. These outcomes included indicators for the efficacy of TB therapy were TB mortality rate, TB rate among the household contacts of the Index TB cases, TB treatment failure, HIV associated TB death rate, TB defaulter rate, and new TB smear positive cases as indicated in Table 1 to Table 8, and Figure 1 to Figure 4, our published data elsewhere [32]. Analysis of variance (ANOVA) and Turkey’s tests for post-hoc analysis with a type I error rate of 0.05 were used to compare the means of the pertinent variables. Regression models and canonical discriminant analysis (CDA) were used to examine the associations between trends in TB incidence and independent TB predictors [32]. Wilk’s Lambda values were closer to zero during CDA as indicated in Table 6, and Fischer’s linear functions, Eigen values, and Mahalanobis distances were also calculated as shown in Table 5, of the published data [32]. The data was analyzed using SPSS^®^ statistical software, version 23.0 (Chicago, IL, USA), and more information on the analysis can be found in our previously published article, which informs part 1 of this model development paper [32].

### 3.2. Part 2

This study used a qualitative approach at this stage, employing the aggregate complexity theory and ethnographic ideas [34,35]. Complete mental representations of object function were provided by ethnographic principle hypotheses [35]. It has made it simpler for us to comprehend our perspectives, which is the first step in fostering reflexivity (the analysis of one’s own beliefs), a crucial skill for conducting the exhaustive qualitative research required for our study [35]. To comprehend the local language and social customs, ethnography demanded total immersion in the culture. Conversation with participants in their native tongue enhances immersion. Before beginning this investigation, the researchers received training from an ethnographer. This was crucial for developing the question guide, testing it beforehand, and conducting participant interviews [35]. The interconnections between sociocultural, behavioral, interpersonal, and environmental aspects determined the framework during analysis. When it comes to TB-HIV service integration models and variables influencing how well they are perceived, culture has a significant impact on behavior and outcomes [35]. The sample for this qualitative research phase involved 25 health service providers and 29 TB/HIV patients from the five selected health facilities in the O.R Tambo area. The participants were specifically chosen due to their lengthy work histories and direct involvement in the management of TB and/or HIV in the specified facilities. Even with the same frequency of TB and HIV in each of their catchment areas, the five health care facilities were chosen using simple randomization. These findings informing part 2 where published elsewhere as preprint [35].

### 3.3. Part 3

Part 3 was focusing on the perceptions of the patients on TB and HIV integration, about 24 participants and their health facilities surveyed in the study. The study investigated knowledge gaps, potential barriers, and facilitators to TB-HIV integration services in facilities from the patients’ perspective, the researchers found three themes, each with one to four subthemes. The findings of was that patients were unclear about the causal relationship between TB and HIV. They felt that health care providers need to continuously engage in health education to help patients to a better understanding of TB, HIV, and co-infection, especially how they are related. The TB-HIV integration management services in the health facility misunderstood the integration of TB-HIV services in the clinic and believed that TB and HIV services were very rarely provided. The study showed that management of TB-HIV integration needs improvement and that programmes are not linked (partial integration), which drives up treatment costs for patients and complicates the number of visits needed to receive the desired treatment [19]. The study found that nearly half of individuals with a confirmed TB diagnosis did not undergo HIV testing or counselling. This could be due to inconsistent testing and counselling by healthcare professionals. HIV weakens the immune system to opportunistic diseases such as TB, so it has been strongly recommended that TB patients be routinely tested for HIV [19]. The majority of study participants did not receive information about antiretroviral drugs (ARVs), HIV, nutrition, tuberculosis, or other preventive measures, which was considered in the study to indicate TB-HIV integration. Health care worker health education initiatives at TB-HIV were not consistent. The TB-HIV control management programmes emphasizes treatment adherence, particularly awareness of side effects, balanced behavior change in sexual life (condom use, abstinence, fidelity, reduction in number of partners by patient), lifestyle changes (e.g., reduction in smoking and alcohol use), and strategies to support treatment adherence, which may include directly observed treatment [19]. We also investigated whether the facility employed an infection prevention and control (IPC) nurse responsible for implementing policies, knowledge, and education in the facility based on the TB-HIV integration guidelines. The majority of respondents did not know that their facility had an infection control nurse, suggesting that infection control policies are not as effective as they could be and that there are difficulties in implementing successful IPC programmes [19]. We investigated the possible causes of no adherence to treatment protocols. Distances between respondents’ homes and health care facilities that prevented respondents from receiving treatment as planned emerged as one of the reasons. The survey found that the majority of respondents were concerned about the distance to clinics, the condition of roads, and transportation costs, all of which negatively affected their ability to access TB and HIV treatment. Barriers to accessing TB and HIV services included distance and poor infrastructure, particularly in underprivileged populations. Patients recommended improving integrated health services TB-HIV [19]. These findings have been published elsewhere [19].

### 3.4. Validity 

Validity is the extent to which an empirical measure accurately reflects the concept it is intended to measure. For development of this model validity, researchers traced the intermediate results and compare them with observed outcomes. By checking, the simulation model output using various input combinations model was validated. In addition, by also comparing final simulation result with analytic results. PHC nurses and experts in research and model development were consulted during the questionnaire’s preparation to ensure its face validity and content for O.R Tambo District Municipality clinics [34].

### 3.5. Reliability 

Describes the degree to which independent administration of the same instrument or a substantially comparable instrument consistently produces the same or similar results when subjected to the same conditions. To assess model homogeneity, the questionnaire was distributed to five PHC practitioners to fill them [34].

### 3.6. Definition of Operational Concepts

Pneumocystis pneumonia: Pneumocystis pneumonia. Pneumocystis pneumonia (PCP) is a serious infection caused by the fungus Pneumocystis jirovecii. Most people who get PCP have a medical condition that weakens their immune system, like HIV/AIDS, or take medicines (such as corticosteroids) that lower the body’s ability to fight germs and sickness [13].

Simple randomization: It is based on a single sequence of random assignments and it is the technique that maintains complete randomness of the assignment of a subject to a particular group. The most common and basic method of simple randomization is flipping a coin [34].

Analysis of Variance (ANOVA): is a statistical technique to analyze variation in a response variable (continuous random variable) measured under conditions defined by discrete factors (classification variables, often with nominal levels) and allows a comparison of more than two groups at the same time to determine whether a relationship exists between them [34].

Turkey’s test: It is a single-step multiple comparison procedure and statistical test and can be used to find means that are significantly different from each other, named after John Tukey [32].

Wilk’s Lambda: It is a measure of how well each function separates cases into groups. It is equal to the proportion of the total variance in the discriminant scores not explained by differences among the groups. Smaller values of Wilks’ lambda indicate the greater discriminatory ability of the function [32].

Eigenvalues: They are the special set of scalars associated with the system of linear equations. It is mostly used in matrix equations. ‘Eigen’ is a German word that means ‘proper’ or ‘characteristic’. Therefore, the term eigenvalue can be termed as characteristic value, characteristic root, proper values, or latent roots as well [32].

Canonical discrimination Analysis (CDA): It builds a predictive model for group membership. One primary purpose of CDA is to separate classes (populations) in a lower dimensional discriminant space [32].

Mahalanobis distance: It is the distance of the test point from the center of mass divided by the width of the ellipsoid in the direction of the test point. It can be used to determine whether a sample is an outlier, whether a process is in control, or whether a sample is a member of a group or not [32].

Post-hoc analysis: Post hoc in Latin means ‘after this’. Simply put, a posthoc analysis refers to a statistical analysis specified after a study has been concluded and the data collected. A posthoc test is done to identify exactly which groups differ from each other [32].

Regression Analysis: Regression analysis is a powerful statistical method that allows you to examine the relationship between two or more variables of interest [32].

Model: A model is a symbolic depiction of reality, which provides a schematic representation of some relationships among phenomena and uses symbols or diagrams to represent an idea [34]. For the purpose of this study, a model refers to a schematic presentation of the integration of TB and HIV service in O.R Tambo, Eastern Cape.

Context: The context represents the area where the action takes place. The TB and HIV services are provided in the Primary Health care facilities in the five clinics of O.R Tambo District Municipality, Eastern Province [32].

## 4. Results 

### 4.1. Some Key Findings from the Original Investigations (Parts 1, 2, and 3)

Quantitative data from part 1: There were 298 cases of tuberculosis (TB) for every 100,000 people over the course of a five-year period, according to an examination of 62,400 records of TB notices from 2009 to 2013. By the time the evaluation was finished, the incidence of TB had fallen by 79.70% from the baseline data from 2009. Multiple linear regression analysis revealed a strong and independent correlation between the decline in TB incidence and PHC expenditure per capita as well as the cost per patient day equivalent (PDE). In addition, compared to places with less socioeconomic disadvantage, the death rates from HIV-associated TB were significantly higher Table 1 to Table 8 and Figure 1 to Figure 4, as can be seen in our earlier article published [32].

Qualitative data from Part 2 revealed that in a total of 54 participants in O.R Tambo District Municipality clinics, 39 (72.2%) reported that TB and HIV services were partially integrated while 15 (27.8%) participants reported that TB/HIV services were fully integrated (Figure 1 to 4) [35] preprint. By partially integrated is when tuberculosis and HIV/AIDS service delivery points are in the same health facility, but some HIV/AIDS services are provided in TB clinic, and some TB services are provided in HIV/AIDS clinic. To get the full range of TB and HIV/AIDS services, the co-infected patients must still go to two different clinics run by different staff members. However, when TB and HIV/AIDS services are delivered at the same delivery location in the healthcare facility by the same team, TB and HIV/AIDS services are fully integrated [35] preprint. Qualitative data from part 3 observed that 29 patients analyzed using qualitative content analysis and presented from the themes were lack of health education about TB and HIV; an inadequate counselling for HIV and the antiretroviral drugs (ARVs); and poor quality of services provided by the healthcare facilities. Thus, these findings imply that the O.R Tambo district’s TB-HIV integration needs to improve with immediate effect [19]. Therefore, we identified these concepts that inform results of published manuscripts 1, 2 and 3: inputs are actions that must be taken in order to fully integrate TB and HIV. Policies and guidelines (budget, TB drugs, ART, TB-HIV educational programmes, staffing, challenges with infrastructure, public-private partnership, patient flow, and TB and ART suppliers) are among these activities. Processes—input execution; outcomes: intensified case finding, improved access to care, intervention, and TB and HIV prevention are measurable results. Effects of integration (the activities performed have an impact on the efficiency and effectiveness of the TB and HIV programmes; they are fair and responsive) for creating an integration model, as shown in Figure 1.

### 4.2. Model Synthesis

Information extraction from quantitative and qualitative data allowed the key barriers to the full integration of the TB and HIV control and implementation models in the primary healthcare facilities of the O.R Tambo District Municipality to be identified. It has a detrimental impact on patient management since health care professionals are required to be trained in how to support and care for TB and HIV patients in the community. Competent health care staff members can and will inform patients’ family in the neighbourhood about TB and HIV. This will improve patient compliance with therapy, help them comprehend the connection between TB and HIV, be aware of potential drug adverse effects, and be familiar with TB signs and symptoms.

### 4.3. Derivation

TB predictors and their numerous deprivation indices, as well they affect the most susceptible populations in relation to TB and HIV, were identified using the results from quantitative data. Funding can be directed toward the most in need by keeping in mind that the most impoverished people are most affected by TB. TB predictors, multiple deprivation indices, and their effects on the most disadvantaged in connection to TB and HIV were identified using the findings from quantitative data. Knowing that TB mostly affects the most poor sector of the community can help with the distribution of budget to the most in need.

### 4.4. Deductive Reasoning

Deductive reasoning was necessary to operationalize the paradigm that was constructed. Deductive reasoning involves moving from broad to specific conclusions. Deductive reasoning can be used by the researcher to move from a broad theoretical understanding to a testable hypothesis [34]. The TB-HIV service integration model should be abstract in order to direct it toward deductive reasoning and migrate from abstract notion to well-defined strategy before being implemented in PHC facilities.

### 4.5. Inductive Reasoning

As defined by Brink et al. [34] as when special circumstances are combined to produce the general premise, inductive reasoning is a sort of analytical reasoning used when data need to be extended from a small sample to a larger sample. Starting with an observation, patterns in the observation are then found. In order to identify the integration as the study’s core concept, the model generation approach employed in this study’s data analysis from the focus group discussion [19] must use inductive reasoning.

### 4.6. Concept Analysis

Concept analysis is a procedure that enables investigation of the features or characteristics of the concept, according to Brink et al. [34]. Concepts that will be employed in the study were identified, described, and categorized throughout this phase of the research process [34]. Concept analysis is defined as a process that allows for the investigation of the features or attributes of the concept. Throughout this stage of the research process, concepts that will be used in the study were named, described, and organized. Concept analysis includes the procedure by which the traits necessary to the meaning of the concepts are identified, in addition to identifying and clarifying the concepts and variables upon which the model is based. In order to identify whether an idea is the major concept or a sub-concept, Brink et al. [34] suggest that it is crucial to ascertain the concept’s nature and organizational structure. Throughout the idea analysis, concepts with related meanings and others with different meanings but ties to the primary concept will surface [34]. Analysis of the related concepts is required to ascertain how the linked concepts relate to and are related to the central ideas of the model. Defining the integration idea as well as its associated concepts, key components, antecedents, and consequences is part of the concept analysis pattern employed in this work.

### 4.7. The model Validation Phase 

Model validation seeks to confirm the model’s applicability to the integration of TB, HIV, and patient care at the PHC facilities of the O.R Tambo District Municipality.

The model’s efficacy was evaluated through validation, which also established if the model addresses the research issue and the study’s goals. The model’s validation took into account the concepts’ clarity, its range of application, the extension of its use, and its logical progression.

### 4.8. Model Validation 

A model was validated in order to assess its effectiveness. This then determines whether the model satisfies the objectives of the study and offers a solution to the research problem. In this study, the model was validated to see whether the study’s objectives were met. The questionnaires and the model’s structure were delivered to PHC nurses and model development experts from Walter Sisulu University in order to validate the model. Throughout model validation, the primary objective of the model was to demonstrate its applicability to the integration of TB, HIV, and patient care at the PHC facilities of the O.R Tambo District Municipality. Via validation, it was determined whether the model satisfied the objectives of the study and the research issue as well as how well it performed.

## 5. Discussion: Creation of the Model

Our goal was to develop an integration model that would be practical, affordable, and able to handle the demands and challenges of the participating institutions while also addressing the needs and challenges of the environment. Key informant viewpoints and a review of the literature on integration made it clear that our approach had to enable internal variation between sites in addition to being applicable to the setting of the larger health system. Our technique did, however, incorporate broad generic elements that permitted some site-specific customization in recognition of the fact that “one size doesn’t fit all” [23,24]. In addition to the lack of standardized metrics to quantify integration [17,18,19], it is difficult to evaluate the advantages and disadvantages of integration efforts across various settings because the term “integration” has been conceptualized in a number of ways [16]. As opposed to two extremes of integrated/not integrated, it has been described as a “spectrum” [25] or a “continuum” [26]. Moreover, integration has been divided into “full” (providing a wide range of services on-site) and “partial” (relating to off-site services) forms [27,28]. At the facility level, the provider, the customer, or both may start or lead it. It is also important to establish the direction of integration, i.e., which programs or services are being integrated with which new or existing services? The completely integrated model for HIV-TB services was adopted in South Africa. However, in line with the results of our study, O.R Tambo District Municipality PHCs produced a partial integrated service model. They were used in this model as a reference to implement a full integration model. Processes and inputs must be improved in order to achieve a full integration, which will have an effect on how TB and HIV control are managed. Both programs will be responsive, and patients who are co-infected with TB and HIV will have equal access to services (outcome and integration effects). These services were included in our approach at the provider and facility levels. The latter had a single provider providing a range of services, whereas the former involved procedures to support internal referral, either during the same visit or a subsequent one. Measures to maintain solid ties across services were also put in place to improve coverage and continuity of care. Throughout the model’s development, a few issues showed up like the inclusion of the community but intentionally the study was only focusing among health care workers and patients, however with a health care system that is already overburdened and has a high staff turnover rate. In addition, health care workers focused on how to ensure true community involvement, build capacity without unnecessarily depending on training, and improve service quality. Reviewing the research on potential obstacles to the efficacy of integrated programs revealed additional difficulties. Such obstacles could exist at the facility and health system levels and include a lack of specific instructions on which service is to be integrated in which department and how; insufficient supervisory on integrated training; and stock-outs of TB resources including drugs and ARVs supplier [29,30]. If clients believe that integrated services require lengthy wait times, poor privacy, and little opportunity for questioning during consultations, integration may be less beneficial from their perspective [31].

### 5.1. Inputs 

According to earlier research, policy and guidelines are necessary for integrating TB-HIV care and preparing healthcare professionals for implementation [13]. The budget/healthcare expenses (expenditure per capita, spending per day equivalent and local government expenditure) for integrating have been highlighted in other studies [12]. Studies by WHO [13], UNAIDS [14] and others have shown that TB-HIV education programs have a significant impact on full integration because the two diseases are epidemiologically linked. Patients and communities should be aware that people with HIV infection and untreated latent TB infection are much more likely than people without HIV infection to develop TB disease in their lifetimes. To empower patients and encourage their involvement in the fight against tuberculosis (TB), health education is crucial [15]. Health education initiatives are incorporated into primary health care (PHC) services in South Africa, although they are insufficient [19]. Similar numbers of PNs were employed per 100,000 OPD patients at each facility. In contrast, all clinics had a CHW shortage, which jeopardized appropriate family coverage. As a result, the number of staff members was solely considered a community-level integration constraint. A 2012 South African study that interviewed 29 health managers and NGOs also brought attention to staffing issues. The study found that human resource capacity was a significant barrier at the community level, reflecting the rise in HIV-associated TB cases, financial constraints that made it difficult to create new positions, and issues with hiring and retaining staff [16]. Three to four consultation rooms are needed for integrated HIV-TB care, according to other research conducted elsewhere [13]. Despite having few consultation rooms, 90% of the clinics felt that they lacked enough of them to offer integrated HIV-TB services. Infrastructure seemed to be a major barrier to integration; a related conclusion in the 2014 joint evaluation noted infrastructure issues impeding efficient TB infection control [12]. The historical mechanism of providing TB-HIV services as vertical services located in various locations at facility level is an obstacle for the provision of integrated TB-HIV services, according to one explanation, which points to difficulties reorganizing the location of the services to enhance integration [16]. In Nairobi, Kenya, Chakaya and colleagues investigated getting private healthcare providers to treat TB patients in accordance with national criteria and to report cases to the government [20]. The authors discussed a number of results, including the proportion of patients with private sector diagnoses who were registered with the national program, the proportion of patients who underwent HIV testing (across both sectors), and the proportion of patients who started treatment for both TB and HIV.

### 5.2. Processes 

The movement of TB-HIV patients through a medical facility is known as patient flow. Improvement of patient flow between services and facilities in ways that can benefit all programs including TB and HIV programmes. Individuals who have HIV-associated tuberculosis (TB) frequently have their TB and HIV care provided through distinct vertical programs with little coordination. Such “siloed” (stand-alone) methods are linked to diagnostic and therapeutic hold-ups, which increase needless morbidity and mortality [15,16,17]. National efforts to manage the TB pandemic would be compromised by poor IPC. In addition, PLWHIV are more likely than the general population to get TB infection and die from it. In order to prevent nosocomial transmission of TB, fully integrated HIV-TB services must therefore implement excellent TB infection management, as was demonstrated in the Extensively drug-resistant TB (XDR-TB) outbreak at Tugela Ferry, KwaZulu-Natal, in 2006 [24]. The Interim Policy on Collaboration TB/HIV Activities was published by the World Health Organization (WHO) in 2015. In the policy, efforts for persons with HIV (PLWH) include infection control, increased case discovery, and isoniazid preventative therapy (IPT). Activities for TB patients included antiretroviral therapy (ART), care and support, cotrimoxazole preventative therapy (CPT), HIV counselling and testing, and prevention messages. Targets of the WHO Global Plan to End TB have not been met, despite significant advancements in implementation. Even though a straightforward TB symptom test may have been successfully incorporated, studies conducted elsewhere found that HIV patients were not routinely screened for TB [24,25,26]. Despite the fact that the median CD4 count was 336 cells/mm^3^ (greater than the median CD4 level of 111 cells/mm^3^ observed in large ART cohorts in sub-Saharan Africa), those newly diagnosed with HIV were not provided IPT, even though most may have been eligible. Thirdly, inadequate TB screening procedures were used for HIV patients [27,28,29]. To improve full integration as other research have noted the phenomenon, there should be interlinkage among TB-HIV services and all pertinent units [30,31].

### 5.3. Outcome 

In order to prevent and control TB among people living with HIV, the World Health Organization supports for the integration of HIV-tuberculosis (TB) services and suggests intensive case finding (ICF), isoniazid preventative therapy (IPT), and infection control (the “Three I’s”) [1,2,3]. Full integration has been linked to TB-HIV outcomes such as TB mortality, TB case finding index, TB treatment failure, HIV associated TB death, TB defaulter, and TB smear positive cases, according to previous studies [5,6]. Guidelines and policies supportive of the integration of TB-HIV services (TB, HIV, TB-HIV integrated, and PMTCT) have been developed by the South African Department of Health (S.A Department of Health) [4]. Facilities or PHCs providing integrated TB/HIV services may be able to lower mortality by earlier identification of TB and HIV patients and earlier commencement on both TB therapy and ART. The facility personnel adopted various TB/HIV integration improvements during TB reach and mentors that may have improved TB screening and initiation among HIV patients. For example, taking more proactive steps to identify HIV patients with TB symptoms and shifting TB medications to the HIV clinic. Improvements were also made in genexpert, treatment adherence, patient retention, and comprehensive quality infection control [16,17]. In investigations of groups of HIV-positive patients, heightened case finding with microbiological (sputum smear or culture) research in all patients regardless of symptoms led to the discovery of an additional four cases per 100 people screened. This result is in line with the discovery that when actively screened, sizable portions of HIV-positive people have sub-clinical tuberculosis [18]. 

### 5.4. Integration Effects

As previous research have shown [19,20], improved access, resource utilization, and efficiency would lead to improved TB treatment results. Other concurrent TB or HIV-specific therapies, however, may also result in better results. Consequently, good TB treatment results are then a representation of the influence of all the treatments when TB/HIV integration is offered as part of a national program-improvement approach, as studies in Ghana found [17]. Moreover, it is anticipated that improved access to more thorough care, efficacy, efficiency, responsiveness, and address equity will be connected with increasing integration of TB-HIV care [17]. The main goal of TB-HIV integration is to alter the way that multidisciplinary teams collaborate and deliver care [18]. This is expected to promote continuity, increase coordination, and put the patient first [19].

Intelligibility of the TB and HIV services integration model’s structure.

According to the validation’s results, many respondents thought the model’s name was appropriate and that the integration model for TB, HIV, and patient services was well defined. The TB, HIV, and patient services integration model’s goals are clearly stated, and has showed that the principles utilized in it are straightforward and that it will fulfil its purpose. The model’s context is well stated, according to the respondents. The results also showed that the model’s interactions are understood with structural clarity and consistency, and that they are well represented graphically for visual presentation. The interrelationships between the concepts used in the model are also very well interwoven.

### 5.5. Building Capacity through Instruction and Guidance

A crucial part of the approach is increasing healthcare providers’ ability to provide integrated services. Building capacity served as the foundation for initiatives aimed at bolstering health systems and interventions intended to raise the calibre of certain services. Topics related to TB and HIV were discussed in each case from the standpoint of integrated services. It is critical that training cover “systemic” topics including ways to improve referral systems, monitoring and evaluation (M&E), and record keeping. Sessions should be held on-site, at the providers’ facilities, to boost attendance. Together with community members, they also included non-clinical personnel like clinic receptionists and security guards.

## 6. Conclusions and Recommendations

While taking into account the local context, certain factors were identified as potential barriers to the integration of HIV-TB services. These include input and process factors that have an impact on partial integration, while outcome factors that strengthen integration effects help the system become efficient, effective, and responsive to the needs of the underprivileged population. Improvements in TB and Human Immunodeficiency Virus (HIV) and patients services Integration will be required by addressing the systemic issues affecting health service delivery, including strengthening supervision and mentorship, training, medicine supply, and strengthened designed model for quality care in TB-HIV Integration of Services.

## Figures and Tables

**Figure 1 ijerph-20-05977-f001:**
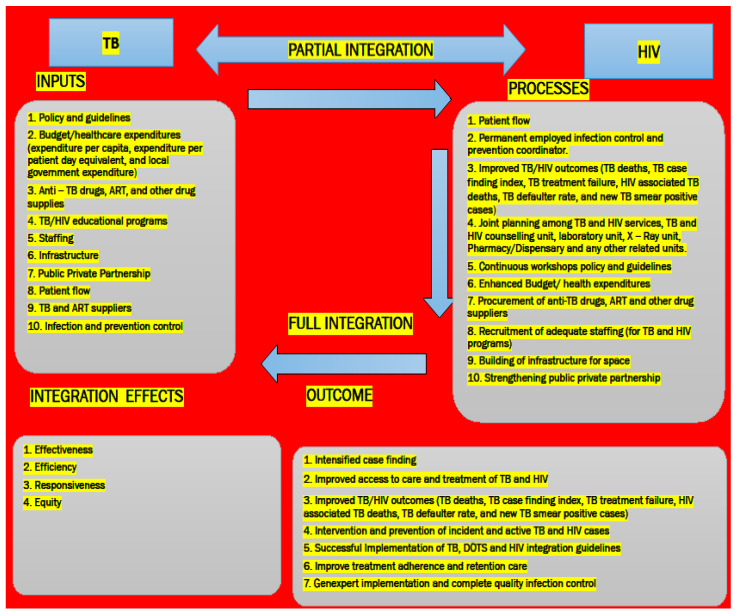
Model of integration produced for the O.R Tambo District Municipality Integration conceptualized.

## Data Availability

The data presented in this study are available on request from the corresponding author. Full data are not publicly available due to privacy restrictions.

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
