# Peer review of "Developing a Model for Integrating of Tuberculosis, Human Immunodeficiency Virus and Primary Healthcare Services in Oliver Reginald (O.R) Tambo District, Eastern Cape, South Africa"

_ijerph, 2023, doi:10.3390/ijerph20115977_

Round 1

Reviewer 1 Report

There are some major issues for this manuscript. Lack of transparent analysis is the main drawback of this paper. Thought explain in the text about the secondary analysis, the authors fail to provide it in the form of figures or table. Results are not presented clearly. Moreover, methods were not thoroughly described. For example, all the quantitative assessment, part 1, the data analysed in the study were obtained from range of sources but fail to provide the details of these data and how these data are compared and analyzed. What are the baseline data for this analysis and lot more.

If this is a research article, please avoid the term “talk” about in line no 59.

In line no 91-102, please provide figures or a table for the calculation. Please show the data analysis in the form of table or figure

In line no 135-136, please provide the exact number of participants for the study and please provide accurate data.

In line no 154, please provide the data.

Line no 171, please indicate how many respondents.

Line no 258-264, please represent this in the form of figure.

Author Response

Dear Editor 

Thank you for the insightful comments and we have responded to your comments. 

Reviewer 2 Report

The manuscript constitutes substantial research, but some amendments are needed:

1. The contribution of the study to the field is not clearly defined. Please, elaborate on this. Specifically, elaborate on the need for integrating the model.

2. You are referring to various statistical methods, but the results are not presented with figures and graphs. An illustration of your results could add up to your research value.

3. A lot of parts stand out in your methodology, but it seems that they are not integrated into a robust framework. Please, present your methodology in a sequence of steps.

4. Although some numeric results are presented, there is no reference to numeric results in the abstract. Please, add numeric results to stress the value of your research in the abstract.

A slight English proofreading is needed.

Author Response

Dear Editor 

We are thankful for the comments you provided us, please find the attached comments as you have requested. 

Reviewer 3 Report

Dear Authors,

The manuscript is well written, organised and has a nice flow. The topic is novel and really interesting. I enjoy reading it and I believe it will be of a high interest to health care researchers. I have spotted some editing issues that should be addressed to improve the presentation. Figure 1, needs to be redrawn with more bright colours and remove the dark shades , the font  also should be larger.

A valid point that should be defenetly discussed and explained in detail: the integration would welcome co-infected or single infected patients? or both?  if both which prevantion methods should be followed to avoid co-infections

Author Response

Dear Editor 

We appreciate the insigthful comments you have provided us, please find attached responses as advised by you. 
